# Composite Wadding of Down Fibers Encapsulated in Fabrics

**DOI:** 10.3390/ma15082825

**Published:** 2022-04-12

**Authors:** Shu Yang, Yingwen Wang, Binjie Xin

**Affiliations:** School of Textiles and Fashion, Shanghai University of Engineering Science, Shanghai 201620, China; wyw1975193932@163.com (Y.W.); bjxin@sues.edu.cn (B.X.)

**Keywords:** down fiber, melt-blown nonwoven, woven fabric, composite wadding, fractal dimension

## Abstract

Down fiber is one of the most superior materials, with excellent thermal properties, that can be used in bedding, clothing, and so on. Down products are usually encapsulated in fabrics that are more compact and, therefore, impart an anti-drilling performance. In this study, down fibers were encapsulated in polypropylene melt-blown nonwoven fabric, and also in polyester woven cloth, to form two different kinds of composite waddings. The waddings made of down fiber encapsulated in melt-blown nonwoven fabrics have a superior moisture permeability, thermal insulation, and anti-drilling performance, and a slightly inferior air permeability compared to that of waddings made with traditional woven fabrics. The pore fractal dimensions of melt-blown nonwoven fabrics are larger than that of woven fabrics. The relationship between the fractal dimension and performance of waddings explains the difference.

## 1. Introduction

Down fiber is recognized as one of the most superior natural materials, and possesses excellent thermal and elastic properties. It often exists in bedding, clothing, and so on. Many researchers have paid attention to the thermal [1,2,3,4] and fluffiness [5] performance of down fibers, some researchers have focused on the grafting modification of down fibers [6,7,8,9], and few on the application of down fibers [10].

Although down fibers possess excellent thermal insulation properties, due to their fineness and light weight, they are difficult to be controlled during fiber processing, easy to drill out during use, and, consequently, the product appearance and consumer experience are poorly affected. Therefore, down fibers are usually encapsulated in fabrics during process and use, such as woven and knitted fabrics.

Compared to traditional fabrics, melt-blown nonwoven fabric has become a more popular covering fabric in wadding due to its fine pore structure. There has been a large amount of research on the manufacturing technique [11,12,13] and properties [14,15] of melt-blown nonwoven fabrics. In this study, the differences between the waddings of down fiber encapsulated in melt-blown nonwoven fabrics and in woven fabrics were investigated.

At the same time, a fractal approach was applied to explain the difference between the two types of waddings. The fractal approach has shown great potential in describing hierarchical systems and also in modeling the heterogeneous structures of fiber assemblies [16,17,18]. Gao [19,20] studied the fractal dimension of a single down fiber. Yang [21] analyzed correlations between the fractal dimension and properties of down fiber assemblies.

In this paper, two different kinds of down fiber composite waddings were investigated, which were made of down fibers encapsulated in polypropylene melt-blown nonwoven fabric and in polyester woven fabrics. The structural parameters and properties of these two kinds of waddings were investigated, such as their air permeability, moisture permeability, thermal performance, and anti-drilling performance. In order to explain the mechanism behind why these two waddings are different, the pore fractal dimensions of melt-blown nonwoven fabrics and polyester woven fabrics were calculated. The relationship between the pore fractal dimension and properties of the two waddings was also analyzed.

## 2. Materials

### 2.1. Down Fiber

Down fibers were purchased in Haining Sea Snow Leopard Down Ltd., Jiaxing, China. The average length of down cluster is 20–25 mm, and the average fineness of down branch is 19–21 μm.

### 2.2. Melt-Blown Nonwovens

Polypropylene (PP) slices were purchased from Shandong Dawn Polymer Material Co., Ltd. (Yantai, China). The nominal molecular weight is 80,000, the ash content is ≤200 PPM, and the moisture content is ≤0.2%. The polypropylene nonwovens were manufactured by the melt-blown equipment (SH-RBJ), which was produced by Shanghai Sunhoo Automation Equipment Co., Ltd., Shanghai, China. It has a spinning jet, a compressed air heater, and a rotating drum (collector). PP slices were fed to the extruder, where they were melted under inlet/outlet temperature of 120 °C/230 °C, and then fed to the spinning jet with through-put of 0.1 g/min. The velocity of the air at the exit of nozzle was set at a few thousand meters per minute to blow the nonwoven fabric to deposit on the collector, with a collecting distance of 15 cm.

Three different series of melt-blown nonwoven fabrics were acquired with different thickness and gram per square meter, which were numbered as N1, N2, and N3. Parameters of nonwoven fabrics are listed in Table 1.

### 2.3. Polyester Woven Fabric

Polyester woven fabric was purchased from Jichuang Textile Ltd., Jiaxing, China. and was numbered as W#. The basic parameters of woven fabric were tested and summarized in Table 2. The average thickness of samples W# and N2 are the same, so these two series of samples may be compared in the following performance tests. In addition, it can be seen that, with nearly the same thickness, the mass of polyester woven fabrics is much heavier than melt-blown nonwoven fabrics.

## 3. Methods

### 3.1. Preparation of Composite Waddings

Three specifications of polypropylene melt-blown nonwoven fabrics N1, N2, and N3, and a polyester plain cloth W#, were separately cut into two pieces of 30 cm × 30 cm square samples. Then, two pieces were stitched using a slit edge stitching machine to form a bag, filled with down fibers of 5 g, 8 g, and 10 g, and the fourth side was also stitched to seal up the wadding. The flowchart of composite wadding is shown in Figure 1. The composite waddings that encapsulated 5 g down fibers in N1 nonwoven fabric is named N1-5, and the same applies for the other nonwoven fabrics. Parameters of all of the waddings are summarized in Table 3.

Several measurements were applied to characterize the properties of composite waddings. Ten samples were tested in each experiment to acquire accurate results.

### 3.2. Air Permeability

Air permeability performance of composite waddings were tested using YG461D digital fabric air permeability tester (Fangyuan Instrument Ltd., Wenzhou, China). According to standard GB/T5453, testing area was chosen as 20 cm^2^, experimental pressure was set as 100 Pa, and a suitable nozzle was chosen. Measurements were acquired under standard temperature (20 °C) and relative humidity (65%).

### 3.3. Moisture Permeability

Moisture permeability performances of composite waddings were tested using LCK-131 water vapor permeability tester (Fangyuan Instrument Ltd., Wenzhou, China) with a silica gel dryer (Fangyuan Instrument Ltd., Wenzhou, China). According to standards GB/T12704-1991, the moisture mass of fabric can be calculated according to
(1)WVT=(24×Δm)/(s×t)
where *WVT* (g/m × 24 h) is the moisture mass per square meter per day, Δ*m* (g) is the difference between two weights of the samples, *s* (m^2^) is the area of sample, and *t* (h) is experiment time.

### 3.4. Thermal Insulation Property

The experimental instrument adopted was 606D YG (b) plate insulation system (Fangyuan Instrument Ltd., Wenzhou, China), and, according to standards GB/T11048, the test was performed under standard temperature (20 °C) and humidity (65%). The temperatures of the protection plate, test plate, and base plate were all set as 36 °C, preheating time was set as 30 min, and the number of cycles was 5.

### 3.5. Anti-Drilling Performance

Down fiber products in daily use undergo two main forces, which are friction and washing force, so the anti-drilling tests were implemented by simulating these two forces.

**Simulation of friction force** The surface of the samples was cleaned before test. Simulation of friction was realized by the ball milling apparatus (Fangyuan Instrument Ltd., Wenzhou, China). After 30 min of rubbing on fabric surface, down fibers drilled out of samples that were longer than 2 mm were counted by magnifier and tweezers.

**Simulation of washing force** The surface of the samples was cleaned before test. Washing machine (Midea Ltd., Foshan, China) was used to simulate the washing force. Washing power was applied on samples for approximately 30 min. Later, down fibers drilled out of samples that were longer than 2 mm were counted by magnifier and tweezers.

### 3.6. Observation

Samples were characterized by SEM (Hitachi SU-70) (Hitachi Ltd., Tokyo, Japan) after coating with platinum.

### 3.7. Fractal Approach

Fractal geometry, as a tool for studying irregular geometrical morphology, was first proposed by B. B. Mandelbrot [22]. It is applicable for a system that is disordered, complicated, and scrappy, but possesses specific self-similarity. Fractal dimension is a value without a unit to describe the degree of fractal in fractal morphology [23]. There are three definitions of fractal dimension [24,25], which are Hausdorff dimension, box-counting dimension, and similarity dimension.

In this paper, the pore fractal dimensions of melt-blown nonwoven fabric and woven fabric were calculated by box-counting method. This method is according to image analysis of sample images

Before calculation, image pretreatment is needed to obtain clearer images; thus, the calculation results are more accurate. Firstly, grey tonal range of images were extended to cover all, making the contrast between bright and dull more obvious. Then, histograms of images were distributed evenly to achieve a balance. Next, all images were median-filtered to remove noisy points in order to acquire clearer images.

Then, the binarization images of a sample were covered using square boxes with a length of side ε, and the number of square boxes that include at least one white pixel point was counted as N(ε). The fractal dimension can be determined by the slope of a linear fit through the values on a logarithmic plot of the cumulative number of boxes N(ε), versus the reciprocal length of side 1/ε, as illustrated in Formula (2). During the process of calculation, several instances of computer software were needed, such as Matlab, and were used to calculate fractal dimensions. Photoshop was applied to process the images.
(2)$$Dim_{box}(M)=\underset{\varepsilon \to 0}{\lim}\frac{\ln N(\varepsilon )}{\ln(1/\varepsilon )}$$

## 4. Results

### 4.1. Air Permeability

Air permeability is an important attribute of waddings. The results of air permeability for all of the waddings are shown in Figure 2. In the comparison of the waddings that are encapsulated in the same fabric, such as N1-5, N1-8, and N1-10, the air permeability decreases with an increase in the down fiber mass. This is because, when the fiber mass increases, the fibers are more crowded in the wadding, and the pores between fibers become smaller. This indicates that pores between fibers in waddings influence air permeability.

The thickness of the fabric also affects air permeability; this can be seen from the comparison of waddings with the same down fiber mass but encapsulated in different nonwoven fabrics, such as N1-5, N2-5, and N3-5. The results show that the air permeability of the fabric decreases with an increase in the thickness of nonwoven fabrics.

Down fibers with the same mass encapsulated in different fabrics possess a different air permeability, according to the comparison between samples N2- and W-, which have nearly the same fabric thickness. It can be seen that the air permeability of melt-blown nonwoven fabrics–down-fiber composite waddings is lower than that of plain polyester-cloth–down-fiber composite waddings by approximately 10–20%. The reason for this may be due to the more compact fiber assemblies and much finer pore structure in melt-blown nonwoven fabrics.

During the use of waddings, especially in clothing and home textiles, a good air permeability is preferred. From this point of view, less down fiber, thinner fabrics, and woven fabrics seem to be more appropriate.

### 4.2. Moisture Permeability

As another important parameter of composite waddings, the moisture permeability of all of the waddings in both the positive side and opposite side are measured and shown in Figure 3. Firstly, the moisture permeability of both sides of composite waddings are nearly the same, which indicates that the direction of waddings has no influence on the moisture permeability. As can be seen from waddings encapsulated in the same fabrics, the moisture permeability decreases with an increasing fiber mass, most likely due to a reduction in the pore size of waddings. The parameters of nonwoven fabrics also affect the moisture permeability, and it decreases with the increasing thickness of nonwoven fabrics. When the nonwoven fabric is thicker, it is harder for the moisture to go through. In the comparison of waddings encapsulated in N2 and W#, the moisture permeability of nonwoven fabrics is much higher than woven fabrics. Despite the woven fabric having a larger pore size, the wicking effects of fine fibers in melt-blown nonwoven fabrics play a more important role in moisture transfer. A good moisture permeability is preferred during wadding use so that sweat or other moisture can be transferred out quickly.

### 4.3. Thermal Performance

The thermal performance is the most valuable property of down waddings. Three parameters of thermal performance were acquired through thermal measurement, which are the CLO value, heat preservation rate, and heat transfer coefficient, where the CLO value is an engineering unit that describes the thermal insulation properties of fabrics. When the room temperature is 20 °C, the relative humidity does not exceed 50%, the air velocity does not exceed 10 cm/s, and the thermal insulation value or CLO of the clothing worn by a person sitting still or engaging in light mental work to maintain a comfortable state is 1, which is measured by legal measures. The unit is defined as l CLO = 4.30 × 10 K·m·h/J = 155 m·K/W. The results are shown in Figure 4.

Waddings with a high CLO, high heat preservation rate, and low heat transfer coefficient are preferred. As can be seen from Figure 4a,b, both the CLO and heat preservation rate show similar trends in each group. The mass of down fiber influences the CLO and heat preservation rate in a direct proportion. This is because the thermal insulation property of waddings are mainly supported by down fibers.

Meanwhile, the CLO value and heat preservation rate of waddings encapsulated in melt-blown nonwoven fabrics are superior to that of woven fabrics. On the contrary, the heat-transfer coefficient of waddings shows an opposite trend and seems to decrease with the increasing down fiber mass in each group (Figure 4c). Generally speaking, the results in Figure 4 indicate that the thermal insulation property of waddings encapsulated in melt-blown nonwoven fabrics is better due to their finer fiber and pore structure.

### 4.4. Anti-Drilling Performance

The anti-drilling performance is the most concerned property in this study. After applying a rubbing and washing force on the waddings, the numbers of drilling fibers were counted, and the results are shown in Figure 5.

The anti-drilling results after rubbing and washing are very similar. Generally speaking, waddings encapsulated in melt-blown nonwoven fabrics have a more excellent anti-drilling performance than those in woven fabrics, and is nearly triple. This can be explained by the finer pores and more complicated path in melt-blown nonwoven fabrics. In contrast, the fiber mass and fabric thickness have less of an impact on the anti-drilling performance.

In summary, waddings encapsulated in melt-blown nonwoven fabrics have a better moisture permeability, thermal insulation property, and anti-drilling performance, while having some shortage in air permeability.

## 5. Discussion

According to Formula (2), the pore fractal dimension of nonwoven fabrics and woven fabrics can be calculated.

Figure 6a is one of the SEM images of nonwoven fabric N2, Figure 6b is a clearer binarization image of nonwoven fabric N2, and Figure 6c shows the pore fractal dimension of this image.

Figure 7a is one of the SEM images of woven fabric W#, Figure 7b is a clearer binarization image of woven fabric W#, and Figure 7c shows the pore fractal dimension of this image.

After the calculation of ten images for each sample, the average fractal dimensions of nonwoven fabric N2 and woven W# are listed in Table 4.

From Table 4, it can be seen that the fractal dimension of melt-blown nonwoven fabrics is larger than woven W#. Then, the relationship between the pore fractal dimensions of fabrics and wadding properties are established and analyzed.

Figure 8a shows the relationship between pore fractal dimensions and the average air permeability of N2- and W#- wadding. The fractal dimensions are inversely proportional to air permeability. The same trend is reflected in Figure 8d, which shows the relationship between the fractal dimension and drilling number. That is to say, when the fractal dimensions of fabrics are larger, the anti-drilling performance will be greater. Figure 8b shows the relationship between the fractal dimensions and moisture permeability of waddings, and Figure 8c is the relationship between the fractal dimensions and CLO value. They are directly proportional. That means that, when the fractal dimension of the melt-blown nonwoven fabric is larger, the moisture permeability and thermal insulation property are more outstanding.

## 6. Conclusions

The waddings made of down fiber encapsulated in melt-blown nonwoven fabrics have a superior moisture permeability, thermal insulation performance, and anti-drilling performance. However, the air permeability is slightly inferior to that of waddings that are encapsulated in traditional woven fabrics.

The pore fractal dimensions of melt-blown nonwoven fabrics are larger than that of woven fabrics. The relationship between the fractal dimension and performance of waddings reveals the mechanism.

Generally speaking, the fundamental performance of the composite waddings made of down fibers encapsulated in melt-blown nonwoven fabrics is satisfied. Furthermore, the industrial process of melt-blown nonwoven fabrics has a high efficiency and low cost.

## Figures and Tables

**Figure 1 materials-15-02825-f001:**
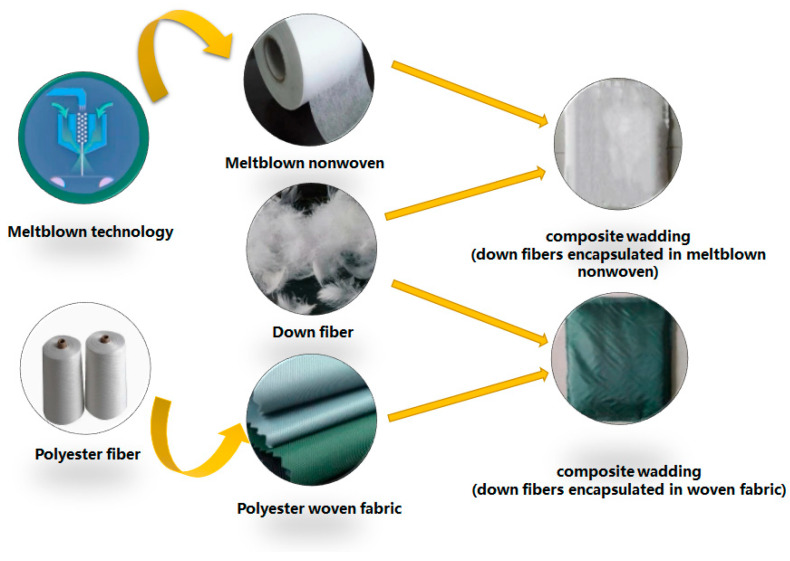
Flowchart of composite waddings.

**Figure 2 materials-15-02825-f002:**
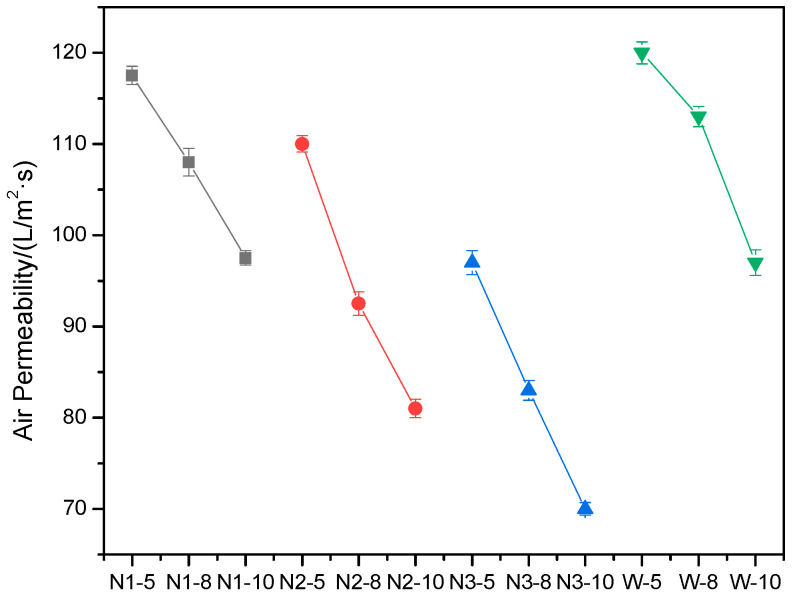
Air permeability results of waddings.

**Figure 3 materials-15-02825-f003:**
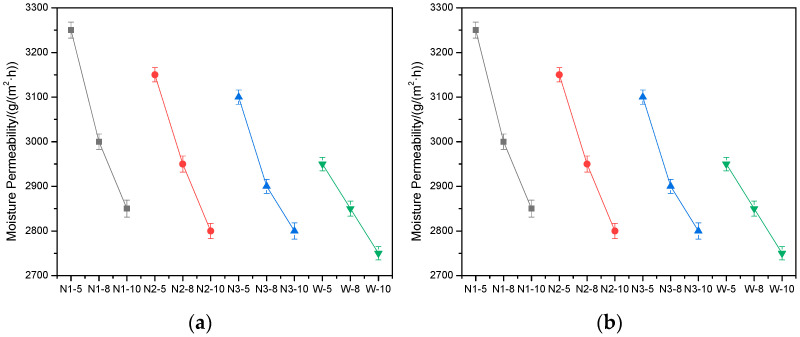
Moisture permeability of composite wadding: (**a**) positive side and (**b**) opposite side.

**Figure 4 materials-15-02825-f004:**
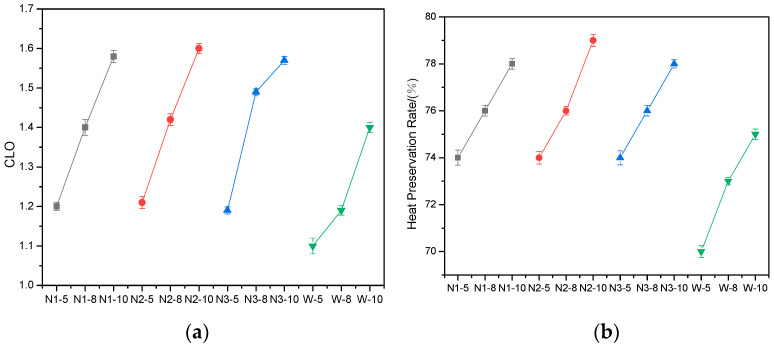
Results of (**a**) CLO values, (**b**) heat preservation rates, and (**c**) heat transfer coefficients of waddings.

**Figure 5 materials-15-02825-f005:**
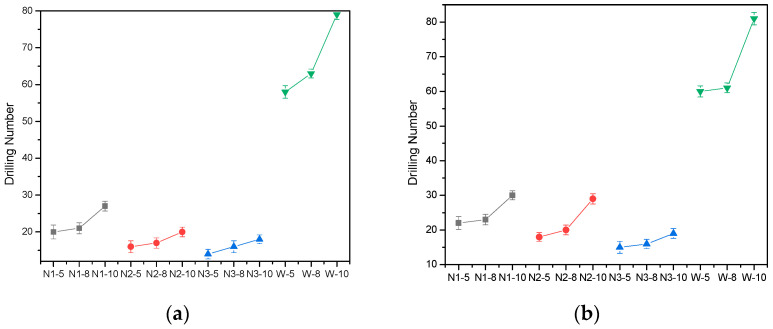
Drilling results of waddings after (**a**) rubbing and (**b**) washing.

**Figure 6 materials-15-02825-f006:**
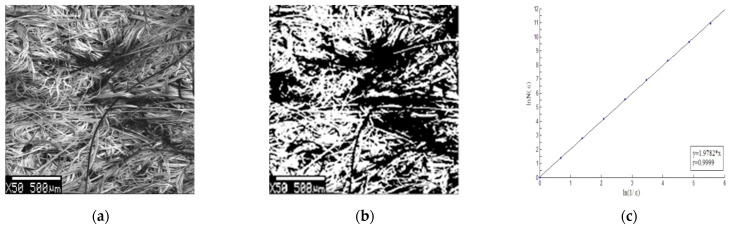
Fractal dimension results of nonwoven fabrics: (**a**) SEM image, (**b**) image after pretreatment, and (**c**) fractal dimension calculation.

**Figure 7 materials-15-02825-f007:**
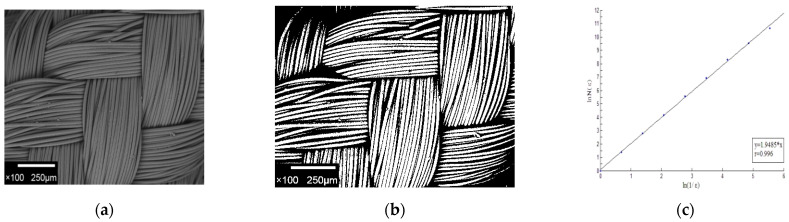
Fractal dimension results of woven fabric: (**a**) SEM image, (**b**) image after pretreatment, and (**c**) fractal dimension calculation.

**Figure 8 materials-15-02825-f008:**
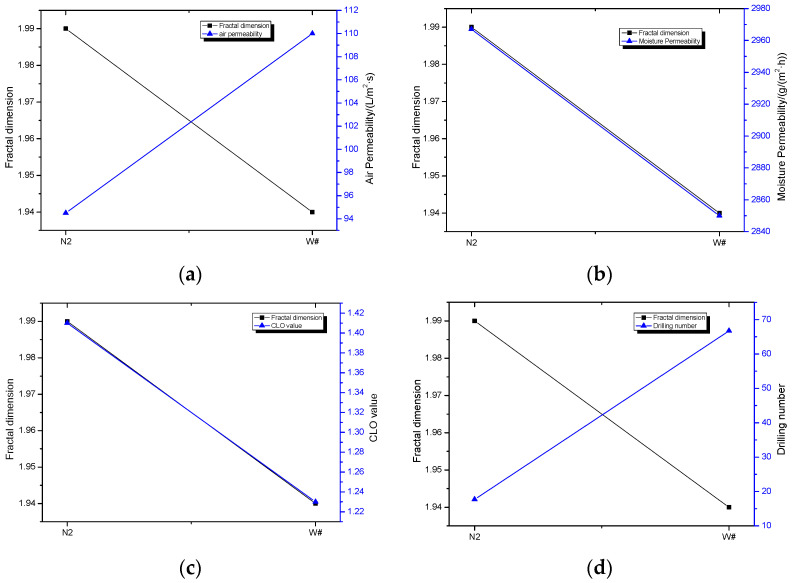
Relationship between pore fractal dimensions and (**a**) air permeability, (**b**) moisture permeability, (**c**) CLO value, and (**d**) drilling number of waddings.

**Table 1 materials-15-02825-t001:** Parameters of melt-blown nonwoven fabrics.

Sample No.	Thickness (mm)	Grammage (g/m^2^)
N1	0.2	24
N2	0.3	33
N3	0.4	48

**Table 2 materials-15-02825-t002:** Basic parameters of polyester fabric.

Sample No.	Weave	Thickness (mm)	Grammage (g/m^2^)	Yarn Size (s)	Density (warp/10 cm × weft/10 cm)
W#	Plain	0.3	150	70 × 70	150 × 150

**Table 3 materials-15-02825-t003:** Parameters of composite waddings.

Sample Name	Covering Materials	Filler (Mass of Down Fiber/g)
N1-5	N1	5
N1-8	N1	8
N1-10	N1	10
N2-5	N2	5
N2-8	N2	8
N2-10	N2	10
N3-5	N3	5
N3-8	N3	8
N3-10	N3	10
W-5	W	5
W-8	W	8
W-10	W	10

**Table 4 materials-15-02825-t004:** Average fractal dimensions of samples.

Sample	Nonwoven N2	Woven W#
Fractal dimension	1.99	1.94

## Data Availability

Not applicable.

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
