# Peer review of "Composite Wadding of Down Fibers Encapsulated in Fabrics"

_materials, 2022, doi:10.3390/ma15082825_

Round 1

Reviewer 1 Report

Composite wadding of down fibers was developed in the study by encapsulating in MB nonwovens. This study has a value to be published in the Materials journal, however, I recommend its publication after considering the following recommendations:

  1. It is mentioned in the heading melt-blown nonwovens were studied. However, in the text, woven fabrics are also used. the heading should be rewritten.
  2. Line 13, 190, 192: The "and" should not be used as the first word of the sentence.
  3. Line21: The reference style should be corrected.
  4. The introduction should be reorganized.
  5. The novelty should be clearly mentioned in the last paragraph of the introduction.
  6. There should be a space between values and units. The manuscript should be checked and corrected.
  7. Line 75: The name of the lab or device should be written. The words like our, their, we should not be used.
  8. The meltblowing procedure and parameters should be written clearly. 
  9. In table 2: what is the unit of density? What is the meaning of 150x150?
  10. Figure 1: It should be drawn again, should be more clear and more compact.
  11. Line 110: What is the cross-sectional area of the test samples. 
  12. The number of the equations should be corrected. 
  13. The air permeability of woven fabric is better than the meltblown sample of N2 (Figure 2). In this study, the authors focused on the meltblown one. What is the advantage of using a meltblown sample? 
  14. Line 182: What is the full name of CLO. It should be written. 
  15. Figure 6: The parameters for SEM are not given. They should be written in the materials and method section.
  16. The difference in fractal dimension is just 1.5% higher (table 4) for the meltblown sample. Is it meaningful to say it is higher? In conclusion, the following sentence "The number of fibers drilling 233 out of melt blown nonwovens is only about 1/4 of that of the woven cloth" was written. The 1/4 is not given in the result part. The calculation should be shown clearly.
  17. For the results section, the obtained results should be supported with literature.

Author Response

On behalf of all the authors, we thank you very much for giving us an opportunity to revise our manuscript, we appreciate you very much for your positive and constructive comments and suggestions on our manuscript. These suggestions help us a lot improve the manuscript.

We have studied your comments carefully and have made revision which marked in red in the paper. 

Attached please find the response to the reviewer’s comments.

Thanks again for your kind suggestions.

Yours,

Shu Yang

Reviewer 2 Report

Dear Authors, Have a look my comments

Author Response

(The authors gave the same response as above.)

Reviewer 3 Report

Please see the attached

Author Response

(The authors gave the same response as above.)

Round 2

Reviewer 2 Report

accept

Author Response

Dear reviewer,

Thank you for your responsible comments and acception.

Yours,

Shu Yang

Reviewer 3 Report

Please see the attached 

Author Response

Dear reviewer,

We are very grateful to your responsible comments for the manuscript. According to your advice, we revised the manuscript . We also made a point-by-point response, please see the attachment.

Yours,

Shu Yang
